# Structural and Biophysical Insights into the Function of the Intrinsically Disordered Myc Oncoprotein

**DOI:** 10.3390/cells9041038

**Published:** 2020-04-22

**Authors:** Marie-Eve Beaulieu, Francisco Castillo, Laura Soucek

**Affiliations:** 1Peptomyc S.L., Edifici Cellex, 08035 Barcelona, Spain; francastillo@ugr.es (F.C.); lsoucek@vhio.net (L.S.); 2Vall d’Hebron Institute of Oncology (VHIO), Edifici Cellex, 08035 Barcelona, Spain; 3Institució Catalana de Recerca i Estudis Avançats (ICREA), 08035 Barcelona, Spain; 4Department of Biochemistry and Molecular Biology, Universitat Autònoma de Barcelona, 08035 Bellaterra, Spain

**Keywords:** Myc, MAX, protein–protein interactions, protein–DNA interactions, intrinsically disordered proteins, biophysics, drug discovery

## Abstract

Myc is a transcription factor driving growth and proliferation of cells and involved in the majority of human tumors. Despite a huge body of literature on this critical oncogene, our understanding of the exact molecular determinants and mechanisms that underlie its function is still surprisingly limited. Indubitably though, its crucial and non-redundant role in cancer biology makes it an attractive target. However, achieving successful clinical Myc inhibition has proven challenging so far, as this nuclear protein is an intrinsically disordered polypeptide devoid of any classical ligand binding pockets. Indeed, Myc only adopts a (partially) folded structure in some contexts and upon interacting with some protein partners, for instance when dimerizing with MAX to bind DNA. Here, we review the cumulative knowledge on Myc structure and biophysics and discuss the implications for its biological function and the development of improved Myc inhibitors. We focus this biophysical walkthrough mainly on the basic region helix–loop–helix leucine zipper motif (bHLHLZ), as it has been the principal target for inhibitory approaches so far.

## 1. Introduction

The MYC transcription factor (also known as c-Myc) regulates the expression of genes controlling the growth and proliferation of cells. MYC was discovered almost 40 years ago as the cellular homolog to *v-myc,* a viral oncogene from an avian myelocytomatosis virus that caused leukemia and sarcoma in chicken (Figure 1) [1,2]. Noticeably, *v-myc* was the first retroviral oncogene to be found in the cell nucleus [3,4,5], which hinted at its potentially direct role in gene regulation. Two additional human paralogs were eventually identified: MYCN (N-Myc) initially observed in neuroblastoma, and MYCL (L-Myc) identified in lung cancer samples [6,7]. Both were later found to be expressed in many additional tissues and tumor types, and the nuclear localization was confirmed for the all Myc family protein members (MYC, MYCL, and MYCN, from now on Myc). MYCN and MYCL display mostly overlapping functions with MYC although with a more limited tissue-specific expression pattern. All Myc proteins are frequently deregulated in human cancers, where their expression level generally correlate with tumor aggressiveness [8,9].

Initial analysis of the MYC sequence hinted, based on the homology with other transcription factors, at the possibility that it would bind to specific DNA sequences; however, when tested, MYC alone displayed only surprisingly weak DNA binding [10]. It was the discovery of MYC’s obligate partner MAX (MYC-associated factor X) [11] that enabled progress towards a better understanding of MYC biology (Figure 1). Indeed, Myc is part of a network of transcription factors, the Proximal MYC Network (PMN). The PMN acts as a central hub in the nucleus, integrating signals from diverse upstream signaling pathways to coordinate and regulate the expression of thousands of target genes necessary for cell cycle progression, arrest/differentiation, and metabolism, among others [7,8,12]. The members of the PMN, of which MAX is the central node, dimerize and bind DNA through a conserved bHLHLZ domain. The interaction of the heterodimers with the Enhancer box (E-box) elements in the promoters of target genes allows them to recruit multiple interacting proteins, leading to transcriptional regulation and active chromatin remodeling [12]. Myc is generally considered a transcriptional activator, recruiting coactivator partners through its TAD domain, although it can also repress the transcription of some target genes [7]. MAX proteins can form homodimers but are devoid of additional functional domain, and thus generate transcriptionally inactive complexes when binding to MYC-target promoters [12]. The heterodimers formed by MAX with the MAX dimerization proteins X (MXD1, MXD3, MXD4), MAX-binding protein MNT and MAX gene-associated protein (MGA), constitute functional antagonists of Myc, shutting down the transcription of Myc-activated target by recruiting corepressor complexes (e.g., in the case of MXD1, 3, and 4, through their SID-mSin3 interacting domain) [12]. 

In most normal cells, MAX is constitutively expressed [13]. In contrast, quiescent cells express low or undetectable Myc levels, which are normally upregulated in response to mitogenic and development signals [7]. Ectopic expression of Myc is sufficient to drive cell growth and proliferation, and it is the relative expression of Myc and MXD that determines the proliferation or differentiation fate of normal cells [12]. Myc displays a short half-life, and its sub-cellular distribution, stability and degradation are finely tuned through multiple post translational modifications (PTMs) [14] and the coordinated interaction with a vast number of cofactors [15]. Unlike many other oncoproteins that promote cellular transformation following activating mutations (e.g., EGFR, Ras or B-Raf), Myc-driven cancers are virtually always due to its overexpression (e.g., following gene amplification) or deregulation (e.g., via tonic signaling from upstream growth pathways, or impaired degradation). Therefore, there is no real opportunity to target any cancer-specific mutant of Myc. Intriguingly, many, and perhaps all tumors appear to become “addicted” to its activity, and even short-term shutdown of its function leads to apoptosis and/or rapid tumor regression [16]. 

Despite the huge body of literature collected since its discovery, our understanding of the molecular determinants underlying Myc function remains surprisingly limited, in part due to the challenges inherent to the study of intrinsically disordered proteins (IDPs). Nonetheless, the demonstration of the relevance of Myc as therapeutic target in cancer [17,18,19] has provided significant drive to overcome the technical hurdles to identify potent and specific inhibitors [20]. In this review, we summarize the structural and biophysical data that have unveiled distinctive features of Myc biology and some hints they provide to target it more efficiently. 

## 2. Functional Organization of the Protein Domains of MYC

The Myc family members share a *N*-terminal transactivation domain (TAD) encompassing highly conserved transcriptional regulation elements called “Myc boxes” (MB), a central region, and a *C*-terminus basic region helix–loop–helix leucine zipper (bHLHLZ) domain (Figure 2) [12]. The bHLHLZ of MYC has been the most studied at the structural level, in part because the MYC transformation activity highly depends on it [10], and also because it is the only domain for which a predicted fold could be inferred from the sequence early on. Coherently, it has also represented the principal “hot spot” for drug design of MYC inhibitors so far [21]. The TAD constitutes a promiscuous region of Myc, interacting with hundreds of protein regulating chromatin remodeling, transcription and Myc stability [15]. Myc is also subject to a variety of PTMs including phosphorylation, ubiquitination, proline-isomerization, acetylation, glycosylation, and SUMOylation. These PTMs can cross-talk and even drive opposite effects in a context-dependent manner [14]. The human MYC polypeptide is 439 amino acids long, while MYCN (464 residues in total) and MYCL (364 residues) differ in the length of the non-conserved sequences [22]. From a pharmacological point of view, the functional redundancy and partially overlapping expression pattern of the three paralogs suggests that an efficient blockade of Myc function might require their simultaneous inhibition. For simplicity, in the following sections we will focus on the MYC sequence numbering.

### 2.1. Identification of the Dimerization and DNA Binding Domain of MYC and MAX

Eukaryotic transcription factors often present HLH and LZ domains, or a combination of both, which fold into multimeric helix bundles and enable the formation of specific heteromultimeric often dimeric—units. This modularity provides means to control function through the differential expression and degradation of each pair member [28]. By homology with other bHLH and LZ-containing transcription factors, the initial studies on MYC suggested that its C-terminus could form homodimers at very high concentration of bacterially over-expressed protein [29], an observation that was however never reproduced in more physiological contexts [11]. In fact, a specific DNA-binding site, the canonical E-box sequence 5′-CACGTG-3′ was rapidly identified by using a chimera consisting of the basic region of MYC attached to the E12 HLH dimerization interface as scaffold [30]. However, no such complex could be detected for a similar chimera containing the MYC bHLHLZ with the remainder of E12. Electrophoretic mobility shift assays (EMSAs) indicated that MYC was much more selective for this E-box sequence than other bHLHLZ proteins like USF or TFE-3, which were more easily displaced by other palindromic CANNTG E-box probes [30]. It became clear that specific DNA-binding by MYC required dimer formation, but MYC alone could not produce soluble and specific dimeric DNA-bound complexes. The search for MYC-interacting partners led to the identification of MAX from co-immunoprecipitation (CoIP) of a GST-MYC bHLHLZ construct with a phage-display library [11]. MAX was rapidly confirmed as the obligate partner of MYC [31,32,33,34]. MAX exists in two isoforms, p22MAX (160 amino acids) and p21MAX (151 amino acids), which results from alternative splicing and lacks 9 amino acids upstream of the basic region (Figure 3) [35]. In contrast to MYC, MAX proteins were able to form homodimers and heterodimers readily binding to E-box probes in biological assays [11,36,37]. 

#### 2.1.1. Specific Heterodimerization by the Leucine Zipper

The structural biology efforts that followed focused on elucidating the molecular determinants of the MYC/MAX specific heterodimerization and DNA-binding. Analytical ultra centrifugation (AUC) as well as circular dichroism (CD) and solution-state nuclear magnetic resonance (NMR) spectroscopies were initially used to characterize synthetic peptides encompassing the LZ of MYC (MYC-LZ) and MAX (MAX-LZ) [39,40,41]. In the absence of MAX, MYC-LZ behaved as a monomer, and displayed only residual helicity, which increased under acidic conditions or at high ionic strength at acidic pH [39]. Already, the observed destabilization of a putative homodimeric MYC-LZ at neutral pH sufficed to explain the behavior of the full-length protein observed in vivo. Classically, the dimerization interface of two-stranded coiled-coils such as LZ domains can be described by the 3–4 repeats of hydrophobic residues at positions *a* and *d* and polar side chains at positions *g* and *e*, according to the heptad repeat numbering system [42,43,44]. In a putative MYC-LZ homodimer, direct interfacial electrostatic repulsions between positions *a* residues Glu410, Glu417 and Arg424 would be highly destabilizing leading to a low probability of existence (Figure 4) [40,41,45].

In contrast, to the MYC-LZ, the MAX-LZ helicity showed no pH dependence. Moreover, MAX displayed spontaneous concentration-dependent cooperative transitions upon thermal denaturation, supporting a folded homodimeric population in solution [45]. Indeed, MAX-LZ is devoid of the interfacial electrostatic repulsions characteristic of the MYC-LZ. Nonetheless, the presence of Met74 and His81 at positions *d* and of Asn78 and Asn92 at positions a weakens the homodimeric LZ compared to an ideal coiled-coil. Accordingly, a MAX mutant construct in which these interfacial residues were mutated to branched hydrophobic ones (i.e., a Val and a Leu, generating MAXVL) evidenced a dramatically increased thermostability as a homodimer [46,47]. Of note, in these experiments, the monomeric MYC-LZ displayed a larger helicity than the dimeric MAX-LZ, possibly due to a comparatively larger number of intra-helical attractive polar interactions within the MYC-LZ [40,45]. 

As expected, spontaneous and preferential heterodimer formation occurred when mixing MYC-LZ and MAX-LZ, and the heterodimers were more stable than the MAX-LZ coiled-coil homodimer [40,41,45]. None of the repulsive electrostatic interactions or suboptimal packing of either MYC or MAX putative homodimers are found in the heterodimer, which instead presents an ideal complementarity (Figure 4) [41]. Interestingly, Lavigne et al. (1998) observed that the stability and helicity of the disulphide-linked heterodimeric MYC/MAX coiled-coil was also modulated by pH, with a maximum helicity around pH 4.5. This highlights the importance of favorable interfacial electrostatic interactions in stabilizing the heterodimer, including one between the ionizable His81 side chain of MAX-LZ and the Glu410 and Glu417 side chains of MYC-LZ buried within the hydrophobic interface of the LZ (Figure 4) [40,41]. Indeed, such pH value theoretically corresponds to the highest relative abundance of negative and positive charges, hence leading to optimized electrostatic interactions at the heterodimer interface. Of note, such high concentration of charged molecules and below-neutral pH values mimic, in a way, the physiological nuclear context [48].

#### 2.1.2. Molecular Recognition and DNA Binding by the bHLHLZ

Following studies extended the characterization to the entire bHLHLZ domains of MYC and MAX and to the full length p21MAX protein. They confirmed, on one hand, the monomeric and intrinsically disordered nature of the MYC bHLHLZ, and, on the other, the monomer-dimer equilibrium of both the MAX bHLHLZ and full-length constructs [46,49]. The Kd(37 °C) for the p21MAX homodimerization in the absence of DNA was found to be in the micromolar range [46,49], a value 10–100 times lower than for the bHLHLZ domain alone [50]. This difference could arise from an electrostatic effect: Indeed, the presence of an acidic *N*-terminal extension in the full-length MAX compared to the bHLHLZ construct could shield an electrostatic repulsion between the positively charged interfacial Basic Region (BR) of the dimeric form. Due to the solubility and favorable biochemical properties of the MAX constructs, the crystal structures of the DNA-bound MAX bHLHLZ were rapidly obtained [51,52]. They confirmed the predicted left-handed, four-helix bundle arrangement formed by the connecting α-helices of the BR and H1, as well as H2 and LZ (Figure 3). Moreover, they exposed for the first time how base-specific contacts are made by bHLH protein modules, more explicitly via the His/Glu/Arg triad of each monomer (corresponding to residues His28/Glu32/Arg36 on p22MAX) [51,52]. The far-UV CD spectra of the full-length p21MAX indicated that the *N*- and *C*-terminal extremities remain unfolded in the absence and in the presence of DNA [50,52]. As expected, a strong increase in the α-helical content of the MAX bHLHLZ and full-length constructs was observed by CD and NMR upon binding to the E-box probes, corresponding to the concomitant folding of the BR. The 1:1 stoichiometry of the dimer:DNA duplex was confirmed by AUC. Differential Scanning Calorimetry (DSC) was used to determine the affinity of the full-length p21MAX and of the bHLHLZ for DNA. Addition of the E-box probe increased the thermogram melting peak of p21MAX by ~7 °C, and the Kd was estimated in the low nanomolar range, in the same order of magnitude as for the bHLHLZ [46,49]. Thus, while the N-terminal and C-terminal extensions of p21MAX compared to the bHLHLZ only contribute to the stability of the homodimer, they are not involved in any significant interactions with DNA. 

In sharp contrast to MAX, the temperature-induced denaturation curve of the MYC bHLHLZ monitored by CD depicted a very shallow transition suggesting a gradual, less cooperative denaturation of the residual secondary structure [53]. Indeed, the MYC bHLHLZ only adopted a more conventional globular fold upon binding to MAX [27] and to DNA [54]. In accordance, solution-state NMR spectroscopy evidenced a poor dispersion of the correlations (cross-peaks) in the ^1^H-^15^N-HSQC (Heteronuclear Single Quantum Coherence) spectra of the ^13^C,^15^N-labelled MYC bHLHLZ at pH 6.8 and 37 °C, supporting the mostly unfolded state of this construct in the absence of MAX [53]. As NMR measurements reflect the average behavior of the ensemble, it is possible that peak broadening occurs due to the timescale of the conformational exchanges and averaging between the many conformers present in the sample in these conditions. 

Recently, Panova et al. used chemical denaturant titrations (CDT)-NMR to improve the detection of rapid exchange events, such as those experienced by the transient conformers of the MYC bHLHLZ [55]. By taking measurements under an array of chemical conditions, the authors deconvoluted the contribution of some sub-populations of the MYC bHLHLZ conformers in the absence of MAX. In native-like conditions, MYC (residues 352–437) showed only half of the expected cross-peaks in the ^1^H-^15^N-HSQC spectra, and none were observed for the LZ (residues 400–437), presumably because of the signal attenuation described above. Changes in the NaCl concentration, pH, and acquisition temperature failed to increase the number of detected peaks. However, titration with increasing guanidinium chloride (GdmCl) led to a more disordered monomeric and less collapsed state, improving the spectra and enabling the detection of all 83 residues amide signals. Surprisingly, persistent helical structure was observed for MYC, even at high concentrations of GdmCl (3.2 M), for up to 30% of the LZ region [55]. From these acquisitions in denaturing concentrations, the authors inferred that the “native” form of the MYC bHLHLZ in the absence of MAX and DNA contains considerable helical structure, in particular around residues 359–373 (BR) and 400–436 (LZ) (Figure 5). Remarkably, the cluster formed by residues 416 to 422 in the LZ (SEEDLLR) is predicted to be 90% helical in the absence of denaturant. Interestingly, this region is responsible for the specific interaction with MAX, and mutations of residues Glu417, Arg423 and Arg424 result in significant homodimerization [17]. In addition, intriguing contacts between residues in the H2-LZ and the H1—different from those observed in the MYC/MAX crystal structure—were observed [54]. Both regions correspond to phosphorylation sites (Ser373 and Thr400) [56], and such phosphorylation likely perturbs the ensemble, contributing to MYC function and/or turnover (Figure 4). Of note, the mutation of Arg367 in the same region is sufficient to allow the homodimerization of the bHLHLZ, presumably by removing interfacial electrostatic repulsions in the dimer and leading to enhanced stability of the HLH hydrophobic core [57]. Overall, in the absence of MAX, MYC constitutes a clear archetype of IDP, displaying a molten-globule like behavior: It is not a pure random coiled polypeptide, but instead a combination of loose cores with persistent secondary structure elements that lacks precise packing. 

When the bHLHLZ of MYC and MAX are mixed together, an important spontaneous increase in α-helical secondary structure is measured by CD spectra [53,58], evidencing how the two partners fold as they bind each other. In fact, the resulting ellipticity at 222 nm is almost exactly twice that observed for the MAX bHLHLZ recorded at half the concentration, indicating that the majority (circa 80%) of the MYC bHLHLZ spontaneously forms an additional quaternary structure with MAX [53]. Similarly, the ^1^H-^15^N-HSQC spectra of an equimolar mixture of ^13^C,^15^N MYC-bHLHLZ and ^12^C,^14^N MAX-bHLHLZ depicts a single set of well dispersed ^1^H-^15^N correlations, confirming that, in these conditions, MYC takes part in one stably folded quaternary complex with MAX. Although in the absence of DNA, the BR of the heterodimer appears highly dynamic (and thus hardly detectable in the NMR spectra), McDuff et al. could confirm the folded status of the heterodimeric HLH (in addition to the LZ) by measuring a net upfield shift of both ^1^H of the δ-methyl groups of the Leu396 on MYC, due to the packing of this side chain on the Phe43 of MAX (ring current effect) (Figure 4) [53]. 

Macek et al. observed that, as previously reported for the LZ domains [40], the interaction of MYC and MAX bHLHLZ depends on the proton concentration: when the pH was lowered from 7.4 to 6.5 in the same buffer (50 mM potassium phosphate, 500 mM ammonium chloride, 1 mM EDTA, and 1 mM DTT), the affinity increased by a 6-fold factor [56]. Moreover, Banerjee et al. [59] showed by fluorescence anisotropy (or fluorescence polarization, FP) that the dimer formation was favored by the presence of some negatively charged ions such as poly-l-glutamic acid. Indeed, while poly-l-lysine did not cause significant effects on dimerization, poly-l-glutamic acid stabilized both the MAX homodimers and the MYC/MAX heterodimers, suggesting that, in the cell nucleus, the presence of the negatively charged DNA or RNA could facilitate the association of these proteins. Comparison of the respective dissociation constants at physiological temperature indicated that the affinity of the MYC bHLHLZ monomer for a TRITC-labelled MAX bHLHLZ monomer is ~3 times greater than the affinity of one MAX bHLHLZ monomer for another [59].

The interaction of MYC with MAX is characterized by a low nanomolar dissociation constant, more appropriate for a tight binding event rather than for a transient complex. Thermodynamically, the exothermic heterodimerization results from the favorable and specific interactions formed upon binding (negative enthalpy) that compensate for the expected entropic penalty of the conformational landscape constriction of the binding partners [60]. On the other hand, the reorganization of the conformational distribution can also contribute a favorable enthalpic effect [61], altogether resulting in the exothermic heterodimerization observed. 

Ecevit et al. [58] showed by stop-flow polarization that the dimerization rate for MYC/MAX is twice as fast as for the MAX/MAX complex formation, and that the MAX/MAX and MYC/MAX dimers recognized DNA at the same rate at physiological temperature. They surmised that the formation of the DNA-bound dimeric complexes occurs via an initial monomeric binding of MAX to DNA that is fostered by coulombic interactions (between the positively charged BR and negatively charged DNA) and by the large capture radius of unfolded protein domains, a faster route compared to a putative initial (hetero)dimer formation followed by DNA binding.

#### 2.1.3. Discrimination between the Cognate and Non-Specific DNA

The mechanism of discrimination between cognate and nonspecific DNA sequences by MAX has been characterized extensively by EMSA, CD, NMR, and mass spectrometry (MS) [49,62,63]. Sauvé et al. analyzed the NMR and CD spectra of MAXVL [47] and revealed that the molecular discrimination involves a combination of conformational selection and induced-fit folding. Indeed, in the absence of DNA, the BR populates partially folded states that can directly fit in the DNA major groove. These partially folded states also position residues in the upper part of the loop in a favorable orientation to establish contacts with the backbone phosphates [47]. Upon DNA binding, an initial conformational selection is followed by a DNA-assisted induced-fit: The adoption of a folded state previously non-accessible in the “apo” form. This step provides the discrimination by stabilizing the α-helical conformation in the presence of cognate DNA, or destabilizing it when non-specific complexes are bound leading to low affinity complexes (approximately 100-fold difference in affinity).

In the case of MYC, the lack of a significant homodimeric population in the absence of DNA, the relatively low solubility at neutral pH and the higher residual helicity of its BR have made it more difficult to understand. Although both the MAX homodimers and MYC/MAX heterodimers were found to recognize preferentially the canonical CACGTG E-box sequence, they can also bind to a variety of non-canonical E-box sequences (CANNTG), albeit with different preferential order [64] In cells, the genomic distribution of the MAX homo- and MYC/MAX heterodimeric complexes overlaps significantly, although not perfectly, possibly due to a differential recognition of the flanking sequences [64,65,66,67] and/or the recruitment to specific locations through interaction with additional protein partners [26,68].

Hu et al. characterized the dimerization and DNA-binding of the heterodimeric complexes with the MYC and MXD1 bHLHLZ by monitoring changes in the FP of a N-terminally labelled MAX-TRITC bHLHLZ construct [69]. The affinity obtained for the MYC/MAX complex binding to a canonical E-box probe was in the low nanomolar range (Keq ~145), similar to that obtained previously by other groups [70]. Interestingly, the free energy differences between the non-specific and specific DNA binding were much larger for the MYC/MAX and MXD1/MAX heterodimers than for the MAX homodimer (although the exact affinity for the non-Ebox probe could not be determined exactly for the heterodimers). This suggests that, within the PMN, MAX likely contributes more to the high affinity binding to DNA, while the heterodimerization partner (e.g., MYC, or MXD1) contributes more importantly to the selective sequence discrimination [69]. Besides, the binding energy changes calculated from the equilibrium constants indicated that the binding of the two halves of the E-box by each monomer in the dimeric complex was nearly independent and displayed little cooperativity. Of note, no significant change in the binding affinity were observed between pH 6.5 and 8.5.

Sammak et al. reported the NMR and crystal structures of MYC/MAX in the absence of DNA [27]. These evidenced the dynamic nature of the BRs although they displayed significant pre-formed α-helical stretches. The use of refined NMR acquisition strategies enabled an almost complete assignment of the MYC and MAX bHLHLZ resonances in the apo heterodimeric complex, with only five missing residues in MYC (Arg357, Thr358, Arg367, Asn368, and Glu369) and five missing residues in MAX (His27, His28, Arg36, Asp37, and His38), all located at the junction between the BR and H1. As for MAXVL, the dimer formation promoted the population of pre-existing, transiently stable helical sub-regions that position some of the residues ready to contact directly DNA (conformational selection). Accordingly, the crystal structures of the apo MYC/MAX dimer display higher B-factor values in the loops (and towards the extremities of the polypeptides). Each BR evidenced distinct conformational properties, the combination of which likely affects the ability of the heterodimer to distinguish non-canonical DNA sequences (e.g., half-site recognition [65] or recognition in different structural contexts). In fact, in all three crystal forms, less of the BR can be observed in MAX than in MYC [27]. Indeed, the highest-resolution structure (Collect 5/6G6K) shows the entire MYC BR, while in the other two, only residues Asn353−His359 are missing. In contrast, the entire BR of MAX is lacking on all the crystals. Even in the structure with the highest resolution, the densities for residues Asp23, Lys24, and Arg25 are still missing, while in the other two, the Asp23−Leu31 and Asp23-Leu32 segments are completely missing. The absence of defined NMR cross-peaks at the junction between H1 and the BR of MYC in the apo form corroborates the highly dynamic nature of this region, where the stabilization of the extension of H1 into the BR results in unfavorable interfacial interactions among the highly charged residues of the BR. Consistently, removal of the BR of both MYC and MAX significantly stabilizes the heterodimer [27].

Nair and Burley [54] resolved the high-resolution (≤2.0 Å) X-ray structure of the E-box-bound MYC/MAX and MXD1/MAX complexes from cross-linked heterodimeric preparations. The MYC/MAX/DNA structure confirmed the critical role of the BR, the loop and the first residue of H2 in contacting DNA. It also corroborated the importance of the conserved His/Glu/Arg triad (positions 359, 363, and 367 in MYC) in specifying the preference for the 5′-CACGTG-3′ motif. Indeed, the H-bond between the His and the central guanine of the E-box dictates the specificity for a purine base at that position; the Glu forms two H-bonds with the adenine and the cytosine at positions 2 and 3 in the E-box sequence (of note, mutation of this Glu to Gln abolishes DNA binding [71]); finally, the Arg forms H-bonds with the central guanine and with the phosphate group between the cytosine and the adenine (first and second positions), thus dictating the identity of the central 5′-CG-3′ dinucleotide. Importantly, this Arg residue differentiates the bHLHLZ proteins binding to the canonical class B E-box (5′-CACGTG-3′) from those binding to the non-canonical class A site (5′-CAGCTG-3′), which instead harbor a hydrophobic residue at that same position. Additional contacts with the phosphate backbone are formed by Lys371 (in the H1) and Lys389 (in the Loop), both conserved at equivalent positions on MAX, and by Lys355, Arg 356 and Lys392 in the BR (specific to MYC). 

Of note, from the structural and functional homology with the transcription factor USF, it was inferred that MYC/MAX could form bivalent heterotetramers that would enable it to control the expression of gene clusters spatially separated [54]. However, FP [69] and atomic force microscopy (AFM) [72] experiments ruled out the existence of a meaningful population of MYC/MAX tetramers in physiological conditions. Instead, AFM measurements revealed how simple heterodimers can equally recognize multiple E-boxes on the same hTERT promoter, which can bend and form a loop that makes the multiple binding events geometrically and sterically possible without the need for tetrameric MYC/MAX species. Moreover, the recent crystal structures and solution-state NMR studies of Sammak and colleagues in the absence and in the presence of DNA showed no specific interaction nor packing that could reflect such tetrameric complex [27].

#### 2.1.4. Post Translational Modifications of the bHLHLZ

The HLH of c-MYC is the target of many post-translation modifications including phosphorylation, acetylation, ubiquitination and sumoylation. The phosphorylation sites on the HLH were identified recently by MS and NMR. They include Thr358 (located in the BR), Ser373 and Thr400 (in the HLH), and Tyr402, Ser405, and Ser437 (which appear relatively less important), all targets of PAK2 (P21 (RAC1) Activated Kinase 2) [73]. Macek and collaborators characterised the impact of phosphorylation at Thr358 and Ser373 on MYC binding to MAX by CD, NMR, and isothermal titration calorimetry (ITC) [56]. While the residual structure of the MYC bHLH-LZ remains relatively unperturbed by this phosphorylation in the absence of MAX, the phosphorylated form is unable to heterodimerize or to bind DNA. In fact, this dramatic effect of phosphorylated Thr358 (pThr358) can be readily deduced from the pre-existing structural information: Thr358 directly contacts the phosphate diester backbone of the E-box. Instead, the phosphorylation of Ser373 strongly impacts heterodimerization by destabilizing the secondary structure and packing with MAX (Figure 4). The pThr358/pSer373 double phosphorylation event on MYC translates into a more favorable binding entropy, but also into a less favorable binding enthalpy, which increases by two orders of magnitude the dissociation constant (6 ± 1 nM vs 376 ± 7 nM) [56]. The loss in enthalpy and the loss in affinity are associated with a reduced helicity of the double phosphorylated complex observed as a lower θ222/θ205 ratio by CD, and as a broadening of the resonances of Thr358 and Ser373 beyond detection in the ^1^H,^15^N-HSQC spectra [74,75,76,77].

The loop region of MYC also contains the ubiquitylation site Lys389 [78]. Interestingly, this loop adopts different conformations in each of the three crystal structures of the apo form [27], all different from the structure of the DNA-bound heterodimer [54]. In contrast, the loop of MAX, which is comparably shorter, presents the same conformation in all the structures of the apo and DNA-bound complex [27]. Interestingly, MAX deletion was found to decrease MYC protein levels and half-life both in normal and pre-malignant B cells [79], perhaps because it increases the exposure of bHLHLZ ubiquitination sites such as Lys389 [78].

Intriguingly, the bHLHLZ domain of MYC/MAX was also found to interact directly with the INI1 subunit of the SWI-SNF complex in a mutually exclusive manner with respect to DNA interaction [80]. SWI-SNF is a chromatin-remodeling complex that uses ATP to alter chromatin structure by repositioning nucleosomes. According to the X-ray structure and NMR spectra, the BR and HLH of MYC undergo significant conformational change upon binding with INI1 (Kd ~44 µM) relative to the apo heterodimer, in particular at the start of H2 (i.e., V393). In MAX, the most significant changes are observed on the H1 and start of H2 residues, while no changes are reported for its BR. The biological significance of this finding remains to be clarified, as the highly context-dependent functionalities of the SWI-SNF complex make it challenging to fully grasp.

### 2.2. Structural Biology of the Myc Boxes and PEST Domains

The *N*-terminal segment of the Myc proteins contains the transactivating domain (TAD) and several highly conserved regions termed Myc Boxes (MB) present in all the family members (with the exception of MBIIIa, which is missing in MYCL). 

#### 2.2.1. MB0 and MBI

The MB0 and MBI were found to bind to Pin1 [81], a prolyl-isomerase that facilitates MYC degradation by catalyzing the Pro63 isomerization in pSer62MYC from trans to cis, to enhance its DNA binding and transcriptional activity, and then catalyzes Pro63 isomerization from cis to trans in the pSer62/Thr58MYC to facilitate dephosphorylation of pSer62, enabling E3 ligases to ubiquitinate MYC. To date, at least 16 ubiquitin E3s have been identified to ubiquitinate MYC. Pin1 was found to interact with unphosphorylated MYC on residues 1-88 (and more precisely on 13–32) through a flexible, unstable and transient complex (a “fuzzy complex”), which Helander et al. have characterized by surface plasmon resonance (SPR) and NMR spectroscopy [81]. Of note, mutation of the Pin1 binding site on MYC appears to stabilize MYC in a transcriptionally less active state. 

The same region of MYCN interacts with Aurora kinase A (AURKA) [82], which stabilizes MYC by preventing its binding to SCF-FBXW7. The interaction of AURKA with MYCN has been described and occurs via residues 28–89, as identified by GST-pull down assays. The crystal structure of the complex between the MYCN 28–89 construct and AURKA shows that residues 61–89 of MYCN bind to a cleft formed by the N- and C-lobes of AURKA, with MYCN residues 76–89 adopting a α-helical conformation [82]. In contrast, residues 28–60 (also important for binding in the biological assays) are not visible on the crystal structure, supporting the high conformational flexibility of the complex. The interaction of MYCN with AURKA was suggested to partially compete with the binding of MYC to the SCF-FBXW7 complex, thus reducing the elongation of the Lys48 linkages in the polyubiquitin chains.

The MBI (residues 44–63) contains the MYC phosphodegron, three highly conserved residues (Thr58, Ser62, and Ser64) that are relevant to MYC function and to its ubiquitin-mediated proteasomal degradation. Indeed, MYC is initially phosphorylated at Ser62, an event that can occur via a number of kinases including CDK1, ERK (extracellular signal-regulated kinase), JNK (c-jun-NH2-kinase) and DYRK2. This Ser62 phosphorylation subsequently promotes Thr58 phosphorylation by GSK3 and the recruitment of the ubiquitin ligases SCF-FBXW7, directing MYC towards degradation [14]. The kinases that phosphorylate Ser64 are yet to be identified. Other potentially important phosphorylation sites in this region include Ser71 and Ser81. 

The MBI was also found to interact with Bin1 (Bridging Integrator 1 protein), a nucleocytoplasmic adaptor protein with tumor suppressor properties [83,84,85]. The exact function of Bin1 in promoting caspase-independent cell death in transformed cells remains elusive to date, but the interaction of Bin1 with MYC can suppress MYC-mediated transactivation and transformation. The binding to Bin1 Src Homology 3 (SH3) domain occurs via a proline-rich poly-proline II (PPII) fragment of MYC (residues 55–68) (Figure 2). The complex has been characterized by intrinsic fluorescence spectroscopy and displayed a Kd in the low micromolar range, in agreement with the intermediate to fast exchange time-scale observed by NMR [24]. In the complex, the side-chain of Thr58 is oriented away from the interacting interface, while the side-chain of Ser62 makes direct contacts with Bin1. Accordingly, a synthetic peptide phosphorylated at position Ser62 was unable to bind to the Bin1C(SH3) domain even at millimolar concentrations, in contrast to the phospho-Thr58 peptide, which displayed the same spectral alterations as those of the unphosphorylated peptide complex. The biological relevance of the interaction of MYC with Bin1 remains to be determined. A recent analysis of the MYC interactome by MS failed to detect Bin1 [15], perhaps because of a relatively lower concentration of unphosphorylated MYC compared to pSer62MYC molecules in the sample.

Residues 98–111 of MYC were found to bind to the TATA Binding Protein (TBP) (Figure 2) [25]. TBP is part of TFIID, a multimeric protein complex including TBP and 13 TBP-associated factors responsible for the assembly of RNA polymerase II at the transcriptional start site of gene promoters. The interaction, characterized by Biolayer Interferometry [86], presented a Kd of ~5 µM at neutral pH. An auxiliary TBF-anchoring motif was also identified from pull-down experiments at residues 115–124 on MYC. 

#### 2.2.2. MBII

The MBII (residues 128–143) largely modulates MYC’s transcriptional activity by promoting the assembly of the transcriptional machinery through interaction with a wide range of transcription factors [12]. These include for instance the transformation/transcription domain-associated protein (TRRAP), a massive atypical kinase lacking a catalytic domain and acting as a large complex scaffold for docking of chromatin remodeling, histone acetyltransferases [12] and ATPase/Helicase TIP proteins, among others. Feris and colleagues characterized the CD spectra of the MYC TAD encompassing residues 1–190 (required for interaction with TRRAPP in coimmunoprecipitation assays) and confirmed that it is largely disordered, albeit with some helical features likely attributable to the MBII [87]. The TRRAPP interacting region itself also displayed a disordered structure by CD, but a significant gain in helicity was observed in MYC and TRRAPP upon addition of ethylene glycol or 2,2,2-trifluoroethanol (TFE). However, only TFE produced a relatively stable interaction between both protein domains. The interaction was also found to rely on a critical tryptophan residue, Trp135 [87]. 

#### 2.2.3. MBIII and MBIV

MBIIIa (residues 188–199), MBIIIb (residues 259–270), and MBIV (residues 304–324) are located next to the PEST domain, a central region rich in proline, glutamic acid, serine and threonine residues. MBIIIa contributes to the transcriptional repression by MYC, possibly via its association with histone deacetylases [88]. In addition, MBIIIb was recently found to interact with the chromatin regulator WDR5. This interaction (with a Kd ~9.3 µM) appears important to direct MYC association with its target genes on the chromatin [26]. The crystal structure of the complex was determined (Figure 2), and MYC was found to bind to a shallow hydrophobic cleft on the surface of WDR5. Importantly, point mutations on MYC that disrupt the interaction with WDR5 also block the interaction of MYC with the majority of its target genes without impacting its expression, localization, interaction with MAX, or binding to naked DNA. 

Finally, the MBIV is also involved in transcriptional activation by MYC, via the interaction with the co-regulators HCFC1 [89] and YY1 [90]. To our knowledge, no structural information is currently available for this region. Phosphorylation rich regions have also been identified between MBIIIa and MBIIIb (Thr247/Thr248/Ser249/Ser250/Ser252) and nearby MBIV (Ser293, and Thr343/Ser344/Ser347/Ser348) [91].

#### 2.2.4. NLS and other Structural Features of MYC

MYC distributes across the cytoplasmic, nuclear and sub-nuclear compartments [92], making it in some ways even more difficult to successfully target by small molecules (SM). Two nuclear localization signals (NLS) have been identified: a strong one, located at MBIV (residues 320–328), and a secondary one juxtaposed to the BR (residues 364–374) [93]. Intriguingly, MYC post-translational regulation at MBI is linked to its specific sub-nuclear localization. Indeed, pSer62MYC is prevalent at the nuclear periphery. Incidentally, Pin1-mediated proline isomerization of pSer62MYC promotes the co-recruitment of the histone acetyltransferase GCN5 to the nuclear pore basket, leading to nearby histone acetylation and gene activation. In contrast, FBW7-mediated ubiquitin-dependent degradation of MYC occurs in the nucleolus [94]. As mentioned before, MYC displays a remarkably short half-life [95] and gets quickly degraded by the ubiquitin-proteasome [92,96,97]. This degradation can take place both in the cytoplasm and in the nucleoli, where MYC and the proteasome co-localize at high MYC expression levels [92]. Intriguingly, the decay rate of transiently-expressed MYC is not constant but biphasic, with an initial rapid phase (half-life of ~20 minutes) dependent on the first 147 residues of its *N*-terminus, and a second slower phase (half-life of ~2 h) dependent on the *C*-terminal bHLHLZ region [98].

A calpain cleavage site identified at residue 298 was found to lead to the formation of a cytosolic product, the “MYC-nick” variant, which inhibits the transcriptional activity of full-length MYC [99]. To our knowledge, no structural studies have been performed on MYC-nick. 

## 3. Lessons Learned from the Direct Myc Inhibitor Screens

Classical approaches for drug discovery have focused on optimizing the interactions between small molecules (SM) and defined ligand binding or enzymatic pockets on globular proteins [100]. However, this approach does not apply to MYC, which lacks binding pockets. Hence, MYC inhibition must rely on the disruption of protein–protein interactions (PPIs). Typically, PPIs encompass relatively large surface areas associated with high binding free energies [101]. However, MYC also lacks significant secondary and tertiary structure when not complexed with one of its biological partners, and therefore does not display well-defined features to be specifically targeted by SM inhibitors. In fact, signaling proteins are often rich in intrinsically disordered regions (IDRs) and these can share a surprisingly high degree of similarity, which renders the identification of selective inhibitors particularly challenging. As a corollary, the screening efforts for such SM should be complemented with additional screenings against homologous or even non-related targets displaying ID regions in order to minimize the risk of off-target toxicity.

Because the MYC/MAX heterodimerization and DNA binding are essential for MYC-driven oncogenesis, the disruption of either interface constitutes an attractive approach, which has been sought by many groups [20,102]. High-throughput screens have yielded many SM candidates able to prevent the dimerization of MYC with MAX. Such molecules were first described by Berg et al. [103], who developed a fluorescence resonance energy transfer (FRET) assay to screen a library of 7000 peptidomimetic compounds, from which they selected five candidates. Four of those were further evaluated using EMSA and ELISA-based assays. The best ones, IIA6B17 and IIA4B20, showed a 2-fold lower IC50 for MYC compared to the homologous transcription factor Jun. Both compounds performed surprisingly well in cell-based assays, demonstrating that despite the limitations of such screening methods (i.e., irreversible formation of conjugated complex, slow maturation of the fluorescent signal and FRET interference from the auto-fluorescent compounds [104]), they could still prove useful when using the appropriate orthogonal controls. Shi et al. eventually developed 120 analogues from IIA6B17 and IIA4B20, and screened them in colony-formation assay, from which they identified mycmycin-1 and mycmycin-2.

The same FRET approach was used to screen a library of 285 planar, hydrophobic SM, also termed “credit card” inhibitors. This yielded 40 shortlisted compounds, which were then tested by EMSA to confirm their ability to disrupt the MYC/MAX/DNA complex formation. Four candidates were further studied in a far-UV CD-based assay monitoring the structural changes accompanying their binding to synthetic MYC and MAX LZ-derived peptides tethered by a thioester ligation [105]. Surprisingly, none of those compounds reduced the helicity of the heterodimeric LZ construct; instead, all maintained or increased the dimer helicity. In a biological assay, the most promising candidate, termed NY2267, appeared more selective towards MYC-transformed cells than Src, Jun, or PI3K-transformed cells. However, it was unable to discriminate between MYC and Jun luciferase reporters. 

Yin et al. based their screening on yeast two-hybrid system (including 32 pairs of bHLHLZ, HLH, HLHLZ, or bLZ interactions), which inherently takes into account the cell-penetration capacity of the screened molecules. Out of 10,000 compounds tested, they identified a number of candidates with acceptable affinity for MYC/MAX, including 10074-G5 and 10058-F4 [106]. The same group then used virtual screening to identify more potent 10058-F4 analogues. Binding to MYC was confirmed by monitoring changes in the intrinsic fluorescence of the compounds upon binding to a synthetic MYC bHLHLZ monomer (residues 353–439) [107]. An initial selection of 48 candidates was eventually narrowed down to four SM showing higher potency (4.6–18 µM). In an optimization effort, the group combined the best attributes of these molecules to generate 17 new analogues. However, this effort did not result in any significant improvement, suggesting that these non-additive properties could very well simply represent the expected behavior of IDP regions capable of binding multiple structures with relatively weak affinities. In fact, two binding sites on the MYC bHLHLZ were identified for 10058-F4 and 10074-G5 through a series of point mutations of a C-terminally labelled MYC bHLHLZ construct and FP analysis of binding to p21MAX and to an E-box probe [108]. Site I for 10058-F4 encompassed residues 402–409 and Site II for 10074-G5 included residues 366–375 (Figure 5). However, competition assays taking advantage of the intrinsic fluorescence of 10058-F4 and 10074-G5 revealed that many non-fluorescent and non-structurally-related inhibitors could also efficiently bind the same sites on MYC [109]. Accordingly, Heller et al. found that the binding of 10058-F4 to the monomeric MYC peptide 402–412 was highly diffuse and mainly driven by entropic contribution [110]. Indeed, the heats of dilution measured by ITC at 25 °C and at 15 °C indicated a low enthalpic contribution to the binding. Titration experiments based on the intrinsic fluorescence of one Tyr residue within the MYC peptide sequence revealed an affinity of 14 µM. The van’t Hoff analysis indicated that the binding free energy (−27.6 ± −8.5 kJ/mol at 25 °C) is dominated by the entropic contribution (−20.7 ± −4.2 kJ/mol), although an enthalpic contribution is also present (−7.0 ± −4.3 kJ/mol). Consistent with previous studies [111], molecular dynamics based on backbone NMR data recorded for the same complex [112] indicated that the binding is diffuse, as MYC (402–412) remains disordered and 10058-F4 is delocalized across several residues.

A third binding site (Site III, residues 375–385) was identified for the non-fluorescent compound 10074-A4 using a CD-based binding assay and shorter MYC bHLHLZ variant sequences [113]. Interestingly, there is very little similarity among the many “low micromolar affinity-displaying hits” identified by all those screening methods. Moreover, Hammoudeh et al. showed that the three sites can be bound simultaneously [113]. Panova et al. used CDT-NMR to build a perturbation/binding map for SM inhibitors, which describes the stabilizing tertiary interaction between residues 360–380 and 400–410 of MYC and 10058-F4, with the hope of improving the selectivity of this scaffold [55]. So far though, the SM deriving from the 10058-F4 appear to display very weak structure–activity relationship [107,114], with most compounds binding to MYC with similar micromolar range affinities. Of note, these sites on MYC are poorly conserved in MYCL and MAX, which in a way provides some basis for specificity (Figure 5).

Kiessling et al. established a FP-based assay to quantify the MYC/MAX bHLHLZ dimer formation and binding to a fluorophore-tagged E-box oligonucleotide, and assessed specificity by comparing it with Jun and C/EBPα homodimers binding to their respective DNA probe [115]. Screening a 17,298-member library identified Mycro1, and a following analysis of structure–activity relationships (SARs) with commercially available derivatives eventually yielded the analog Mycro2. Mycro1 and Mycro2 inhibited DNA binding with IC50s of 30 μM and 23 μM, respectively, while for Jun and C/EBPα complexes the IC50 was >100 μM. The IC50s for MAX homodimeric binding to the same oligonucleotide were only 2–3-fold higher. In a follow-up study, the same group reported the results of a screen of a ~1700-member library of pyrazolo[1,5-α]pyrimidines based on the structures of Mycro1 and Mycro2, from which 5 candidates were selected that also displayed a selectivity factor of 2 for MYC/MAX heterodimers compared to MAX homodimers, revealing a weak specificity of the approach [116]. Hart et al. used the same FP assay to screen a Kröhnke pyridine library, which yielded the KJ-Pyr-9 compound with a Kd for MYC in vitro of 6.5 nM. The KJ-Pyr-9 compound was shown to inhibit the proliferation of various cancer cells with IC50 values from 5 to 10 µM [117]. To our knowledge, no additional study has been reported for this molecule. 

As suggested by the NMR studies from Sammak et al. [27], the plastic nature of MYC-H1 in the apo MYC/MAX dimer may constitute an attractive feature to exploit for targeting MYC with SM, which could trap it in a form unable to bind DNA. In fact, this strategy was the focus of a library of synthetic α-helix mimetics targeting the helical conformation of residues 363–381 of MYC, at the junction with the arginine-rich BR [118]. The initial screen relied on EMSA, then multidimensional NMR spectroscopy of the MYC/^15^N-MAX bHLHLZ was used to characterize more directly the specific binding of the selected candidates. This study concluded that the SM 4da specifically binds to the heterodimer and not to either monomer, and that this binding modifies the heterodimer conformation. Indeed, the structure stabilized by 4da is a heterodimeric form incapable of binding to DNA. SPR was also employed to verify the DNA binding impairment of the heterodimeric MYC/MAX complex following addition of 4da. Cell-based assays indicated an IC50 in the 10–20 μM range in MYC-dependent cancer cells. Of note, the early designs of helical peptidomimetics derived from the H1 of MYC and aimed at disrupting the MYC/MAX heterodimer were also found to directly bind to MYC bHLHLZ, to prevent MYC binding to DNA in EMSAs and to display with potencies of 1–10 µM in cell-based assays [21]. 

Using similar approaches, Chen et al. [119] initially screened 273 compounds by CD, monitoring the spectral change of MYC (370–409) upon addition of the compounds. From this initial screen, seven candidates were selected and further characterized by SPR to confirm their ability to prevent heterodimerization with MAX. The SPR competitive assay consisted of injecting serial concentration of a mixture of the best candidate with the MYC bHLHLZ onto a CM5 chip functionalized with a GST-MAX construct. NMR spectroscopy of unlabeled MYC bHLHLZ indicated that the addition of the SM PKUMDL-YC-1205 to the protein sample caused disappearance of the TOCSY crosspeaks corresponding to Arg372 Hβ-Hγ-Hδ and Ser373 Hα-Hβ. Molecular dynamic simulations also indicated that this SM binds to multiple conformations of the MYC bHLHLZ.

A cell-based protein-fragment complementation assay-high throughput screening platform was implemented by Choi and colleagues. The compound sAJM589 produced a dose-dependent inhibition of proliferation with IC50 values of ~1 μM in multiple cancer cell lines. Binding to MYC was confirmed by Biolayer interferometry using a biotin-tagged MYC bHLHLZ [120]. The precise binding site of this molecule has not been described yet. 

Han et al. reported a combination of in silico screening for a 16 million compounds library, from which they identified 61 hits then tested in EMSA to confirm their ability to disrupt the MYC/MAX/DNA complex [109]. After evaluation of their ability to reduce cell viability and MYC-driven transcription of a reporter gene, one SM, called Min9 was selected for further testing. Evaluation of eight additional analogues of Min9 led eventually to the identification of 5 candidates with suitable properties. A cell line stably expressing a reporter luciferase plasmid was engineered and inoculated in a tumor xenograft mouse model. Three compounds caused a significant reduction in the MYC-driven transcriptional signal in vivo, with compound 361 being the most potent. To establish target engagement by the compound, a cellular thermal shift assay (CETSA) was developed. This assay monitors ligand-induced changes in protein thermal stability from cells treated with the test item, and confirmed that treatment with 361 destabilized MYC. A competition pull-down assay using Biotin-MYC revealed that 361 binds to the same site as 10074-G5 (residues 366–378). Perhaps unsurprisingly, the competitive FP assay indicated an affinity in the micromolar range (3.2 µM). The authors also showed that 361 treatment destabilizes MYC by selectively promoting Thr58 (but not Ser62) phosphorylation, and that this residue on MYC is essential to the mode of action of the compound. Indeed, mutations of Thr58 or Ser62 to Ala both abolished the anti-MYC activity of 361. Protein ligation assay (PLA) confirmed that 361 prevented the MYC/MAX dimer formation in cells even with the Thr58A MYC mutant. Hence, the disruption of the heterodimer appears to directly promote the phosphorylation on Thr58 and consequently leads to enhanced MYC degradation. Hart et al. also identified another compound with similar activity termed 975. Unbiased MS analysis revealed that both compounds are bound to approximately 135 proteins, of which ~38% belong to the MYC interactome.

Recently, our group demonstrated that the Omomyc mini-protein, a MYC dominant negative encompassing the MYC bHLHLZ with four mutations in its LZ [17], constitutes a clinically viable direct Myc inhibitor [121]. The structural characterization of Omomyc confirmed that, as previously published for its transgenic counterpart [17], Omomyc can block Myc function by a three-pronged mode of action: Omomyc can form homodimers and heterodimers with MAX able to bind the E-boxes and displace Myc from the promoters of its target genes, while also forming heterodimers with Myc that are unable to bind DNA. The ^1^H-^15^N-HSQC and CD spectra and thermal denaturation curves of the bHLHLZ constructs evidence that the Omomyc homodimers and the Omomyc/MAX heterodimers display similar affinities in the absence of DNA, with Kd(37 °C) values of ~300 nM, indicating that both species coexist. Upon biding to DNA, the cooperativity of the transition and the melting temperatures of both the Omomyc homodimer and the Omomyc/MAX heterodimer mixtures increase dramatically, and show a transition half-point at ~70 °C. The affinity of those complexes for DNA, although not calculated, appears significantly higher than that of the MAX homodimer for DNA (hence lower than nanomolar Kd value). In contrast, while we detect evidences for the Omomyc/MYC heterodimer formation by NMR (through chemical shifts displacements) and by CD (via the increased helical signal intensity compared to the arithmetic sum of each sample), neither the CD spectra nor an FP assay allowed us to detect any significant binding to DNA of this heterodimeric form. Recently, Demma et al. reported that the binding of MYC, MAX, and of Omomyc to MAX occurs co-translationally, both for homo- and heterodimeric complexes [122]. Moreover, the association of MYC and MAX proteins with the translating ribosomes was efficiently blocked upon treatment with Omomyc. In fact, in Omomyc treated cells, the biotinylated Omomyc and MAX both bound MAX RNA, while the binding of MYC to MAX RNA was ablated.

## 4. Structure–Function Relationship of an Intrinsically Disordered Transcription Factor

Regions of intrinsically disordered structure are highly abundant and occur in approximately one-third of eukaryotic proteins [123,124] and in up to 60–80% of those involved in signal transduction. Under physiological conditions, IDPs exist as dynamic ensembles with minimal defined structure. IDRs (intrinsically disordered regions) constitute functional units acting in a disordered state, in which the polypeptide chain undergoes continuous conformational fluctuations. Their discovery in the 1990s abolished the dogma that, in order to accomplish their biological function, proteins required a folded structure [125]. IDRs are often characterized by a relatively high net charge and low hydrophobicity, owing to their low content in hydrophobic side chains (which often drive a favorable entropic component for proteins to adopt a folded structure), and high content in hydrophilic and charged residues [126]. The conformational status of IDPs ranges from a complete lack of secondary structure to a combination of residual or even significant small segments of secondary structure [127,128]. The extended nature of ID regions and their surface-exposed area makes them suitable to coupled folding-binding reactions with diverse targets through the formation of large interfaces [129]. Instead, other ID segments can constitute flexible linkers participating in the assembly of macromolecular arrays. Their intrinsic flexibility enables them to form complexes of various conformations with a multitude of partners, also termed fuzzy complexes [130]. The high prevalence of hydrophilic residues in the fuzzy complexes is known to enable for a rapid and direct modulation of the interaction pattern through posttranslational modifications [131]. 

These multiple weak and transient interactions by IDPs/IDRs also contribute to the formation of liquid–liquid phase separation (LLPS) [132,133], a fundamental mechanism used by cells to isolate internal material and compartmentalize the intracellular space. LLPS occurs when a supersaturated solution spontaneously separates into two phases, a dense one and a more dilute phase, stably coexisting. The resulting condensates, also called coacervates when referring to oppositely-charged macromolecular species, can form membraneless organelles within cells (for instance Cajal bodies, P-bodies nuclear bodies, and granules). The absence of membrane allows the spontaneous exchange of components in response to alteration in the environment. LLPS plays a crucial role in many important processes, for instance by forming functional centers for biochemical reactions within the cytoplasm and the nucleus. In the nuclei of eukaryotes, LLPS can produce nuclear bodies that maintain, store and modify transcription regulators. Such condensates provide the ideal context for PTMs (e.g., acetylation, sumoylation) that will determine their function [134,135,136]. Examples include nuclear speckles, polyleukemia bodies, nucleolus, histone locus and others [137]. The relevance of LLPS in regulation of gene transcription has been evidenced multiple times [138]. Among others, the dynamic association and dissociation events of nuclear condensates was found to regulate many processes associated with gene expression [139] including chromatin structure organization [140], RNA processing [141] and ribosome biogenesis [142]. 

Another singular aspect of IDPs is linked to their ability to bind multiple partners in dynamic interactions of modest affinities. Because of it, IDPs are especially prone to functional modulation by their concentration [143]. In fact, at high concentrations, the mass-action drive can easily overcome the specificity of their interactions (the difference in energy between the desired and undesired binding interactions), leading to superfluous binding and to consequent toxicity. In line with this, the concentrations of IDPs are typically tightly regulated by transcript clearance, translation rate, and protein degradation. 

The 40 years of intense research on Myc biology and characterization of thousands of its target genes converged into a common conclusion: Myc function is highly cell type and cell context dependent [68]. Paradoxically to their well-defined sequence-specific DNA binding, in some conditions Myc transcription factors can bind to the promoters and intergenic regions of virtually all active genes in a given cell population, as well as to multiple enhancers [144], even beyond E-box-containing regions [145,146]. However, of all the active genes bound by Myc, only a small subset actually directly responds to the changes in Myc levels at its promoter [147]. Indeed Myc proteins have also been described to enhance the overall rate of genome-wide transcription via Myc’s direct impact on RNA polymerase activity [148], leading to a phenomenon termed global amplification. Nevertheless, tumors triggered by Myc and certain Myc-driven biological contexts present characteristic patterns of up-regulated and down-regulated gene subsets that suggest the existence of extra layers of specificities, which cannot simply be explained by the role of Myc as a global amplifier for genes transcribed by RNA polymerase II [147]. Furthermore, many studies have indicated that the actual transcriptional output of specific promoters depends on the Myc levels, suggesting that promoters affinities account for specificity in Myc-dependent gene regulation, and the existence of productive and non-productive modes of DNA binding [146]. By determining the nuclear MYC concentration and occupancy of every promoter at endogenous and exogenous MYC levels by ChIP sequencing, Lorenzin and co-authors [146] estimated the relative affinity of MYC for all MYC bound promoters. The calculation of the concentration of MYC required for half-maximal occupancy of each promoter (EC50) was then used as a measure for the apparent binding affinity. Promoters with low EC50 values (high affinity) encompassed genes involved in RNA binding, translation, ribosomal and biosynthesis, and got saturated at values equivalent to low levels of endogenous nuclear Myc. Conversely, promoters linked to receptor activity, TGF-ß, or hypoxia presented significantly greater EC50 values, which translate into a substantial promoter occupation only when nuclear Myc levels are elevated [146].

The following model can integrate to a certain extent the various structural and biophysical data gathered on MYC to date: Upon increased MYC expression (following growth signal induction), the spontaneous, co-translational and preferential heterodimer formation with MAX occurs at the ribosomes. The nascent MYC/MAX dimers readily translocate into the nuclei, and the superior affinity of the MYC/MAX dimer compared to MAX/MAX homodimer rapidly displaces MAX homodimers from E-boxes. The access to these genomic locations is facilitated by the relative accessibility maintained by the MAX homodimer through its IDP extremities. The increase in the DNA-bound MYC/MAX population contributes to grow the pre-existing condensates or coacervates through a local augmentation in the concentration of fuzzy complexes via the TAD of MYC [149]. As the coregulators of transcription begin to distribute between the fuzzy complex units and the nucleoplasm, the chemical potential of the condensates changes. Early events consequent to this condensates increase would include, for instance, the stimulation of transcriptional elongation by promoting pause release of PolII via localization of P-TFEb (similar to the case observed for other transcription factors [138]) and the phosphorylation of MYC Ser62 by CDKs. The newly synthesized RNAs, in turn, modify the local chemical potential and promote the formation of coacervates. Accordingly, MYC has been found to contribute to the formation of local loops that enhance the availability of polymerase-populated DNA loci through an ATP-dependent process. In this sense, in an energy-rich context, MYC has a dual role of active and passive factor of chromatin regulation [150]. Additional phosphorylation of MYC continuously and spontaneously occurs, as the kinases continue to localize transiently among the fuzzy complexes and coacervates, according to their partition coefficient. Eventually, the prolyl-isomerase reaches the coacervate, leading to MYC phosphorylation on Thr58 by GSK3, which in turn promotes MYC ubiquitination [94]. Concomitantly to these events, phosphorylation of MYC on its bHLHLZ (e.g., Thr400, Thr358, or Ser373) eventually occurs, destabilizing the DNA-bound complex and causing the detachment of MYC from MAX and from DNA (presumably, in this energy rich context, MAX also undergoes phosphorylation by CKII). At intermediate stages, this provides an additional kinetic drive to the MYC/MAX complex to move to other proximal genomic locations. 

Other large protein partners with more defined folds such as WDR5 help direct MYC/MAX to specific genomic locations. This could lead for instance to longer residency time at those sites, effectively stimulating the recruitment of the transcriptional machinery and activation of select gene sets. Both events (increase in ubiquitination and detachment from the DNA) are necessary to export MYC to the cytoplasmic proteasome. As an additional fail-safe measure to ensure rapid termination of MYC signaling, its degradation by the proteasome can also occur in the nucleoplasm, at higher MYC expression levels [92]. This cycle of events leading to MYC degradation happens spontaneously from its very initial translation into a polypeptide, and provides an automatic auto-destruction route to ensure rapid onset and termination of its signaling and transcriptional output, and maintain it under the control of the external growth signals. In this perspective, the higher residual helical fold of MYC bHLHLZ compared to MAX might simply reflect a structural restraint to limit the conformational adaptability to controls, such as its phosphorylation, lowering the entropy of this specific module. In such chain of events, the formation/expansion of the liquid condensates, which depends on the rapid and enthalpy-driven binding of MYC/MAX to DNA, constitutes an important rate-limiting step. It provides an entropic drive and change in the local chemical potential that stimulates the partition of the coregulators of transcription and post-translational modifiers according to their sequence-specified solubility for each phase. This chain reaction offers a robust yet elegant way to coordinate the myriad of biochemical and enzymatic reactions involved in transcriptional activation. The very presence of additional, tissue-specific transcription factors would be expected to similarly modulate the formation, growth and sub-nuclear localization of the phase condensates, thereby fostering a tissue-specific expression program. Moreover, impaired MYC degradation in an energy-rich context would be expected to provoke significant (de-compacting) chromatin remodeling and, if sustained, drive uncontrolled global transcription. 

## 5. Cornering a Slippery Target

The biological relevance and extreme conservation of several Myc structural domains throughout evolution became evident since their early discovery. However, Myc peculiar conformational behavior and elusive—albeit crucial—role in growth and proliferation of cells posed the hardly soluble enigma of how its structure leads to function. The growing body of evidence from molecular and biophysical studies increasingly points at its contribution to the archaic and robust sub-cellular phase separation process as a mean to coordinate the plethora of proteins Myc is found to interact with. This macromolecular and biochemical perspective can also provide relatively simple explanations for the implications of Myc deregulation in tumor progression. 

Currently, the principal clinical approach towards targeted therapies mitigating high MYC expression in cancer has focused on BET bromodomain inhibition. The BET inhibitors (BETi) act by impairing BRD4 regulation of the *myc* promoter, thereby suppressing its expression. However, the efficacy of BETi is limited to some contexts, and does not apply to the majority of tumors where MYC deregulation arises instead from growth factor-independent tonic signaling through MYC or from impaired degradation. Moreover, not all tumors that display *myc* genetic amplification respond to therapy. 

Another recent approach proposed to target Myc indirectly involves targeting Myc interactors that do display an enzymatic active site (e.g., AURKA). However, there are some problems with this approach too: the targeting of kinases, albeit clinically successful in some cases, remains challenging due to the chemical similarity of the ligand binding site across the enzyme family members, and the limited number of H-bonds (critical to define selectivity) that SM can form within the active site. Second, such enzymatic functions are often highly redundant, and cells have the potential to rapidly develop resistance to their inhibition. Instead, Myc itself provides for a much more attractive target, since its function appears to be not redundant. 

However, direct clinical Myc inhibition has been so far unsuccessful. Unfortunately, the few Myc inhibitors that have reached clinical trial have typically displayed high toxicity. It should be noted that these inhibitors have mostly been SMs, which often bind to their target in a diffuse manner, with very limited selectivity and consequent off-target toxicity. Hence, the development of larger molecules, capable of establishing sufficient number of specific interactions to significantly stabilize Myc in an inactive form appears a promising way forward. As these larger molecules can be limited in their cellular diffusion, the use of cell-penetrating and/or nuclear targeting signals is likely required to efficiently reach the desired cellular compartment. In this sense, the optimization of their partition coefficient could help maximize the residency time in the desired compartment. In this sense, the coupling of the dominant-negative and direct inhibitor function of MYC by molecules like the cell-penetrating Omomyc mini-protein certainly constitutes a promising approach [121]. Importantly, when tested in vivo, Myc inhibition by Omomyc showed a wide therapeutic window, and no toxicity has been observed in mice, even after prolonged treatment following systemic expression in all tissues [17,18,19].

Besides this approach, the development of other molecular scaffolds to target Myc are very likely to require high molecular weight protein domains or peptidomimetics. Also, further screening efforts should take into account, whenever possible, a more detailed knowledge of the nuclear environment (e.g., pH and chemical potential of the sub-nuclear condensates) and test binding to Myc in those conditions when possible. 

In our view, the detailed structural characterization of Myc coupled to a broader view bridging physics, chemistry and biology dynamic macromolecular ensembles will provide a richer perspective to grasp the full range of Myc biological activity and pave the way to its successful rational targeting for cancer treatment. 

## Figures and Tables

**Figure 1 cells-09-01038-f001:**
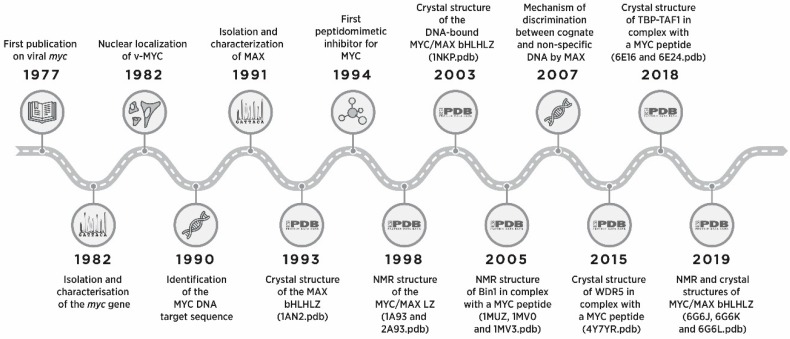
Timeline highlighting relevant achievements related to MYC biology, pharmacology and biophysics.

**Figure 2 cells-09-01038-f002:**
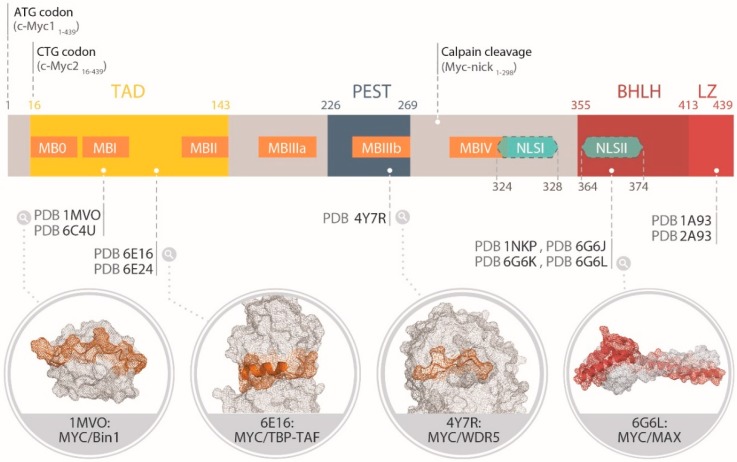
Schematic representation of the modular structure of MYC (UniProtKB code P01106, https://www.uniprot.org/uniprot/P01106). The N-terminal transcription activation domain (TAD, yellow), the central region rich in proline, glutamic acid, serine and tyrosine (PEST, navy blue) and the basic region helix–loop–helix leucine zipper (LZ) domain (bHLHLZ, red) are indicated. The MYC boxes (MB 0-IV) are shown in orange and the nuclear localization sequences (NLS I-II) in cyan. The magnification bubbles show the structural models of MYC fragments (orange for MBs and red for the bHLHLZ) in complex with the interacting partners (light grey) available from the Protein Data Bank (PDB, https://www.rcsb.org) and rendered using Pymol [23]. For the TAD region: 1MVO, in complex with Bin1 [24]; 6E16 in complex with TBP-TAF [25]; 4Y7R in complex with WDR5 [26]. For the bHLHLZ, 6G6L shows the MYC/MAX heterodimer in the absence of DNA [27].

**Figure 3 cells-09-01038-f003:**
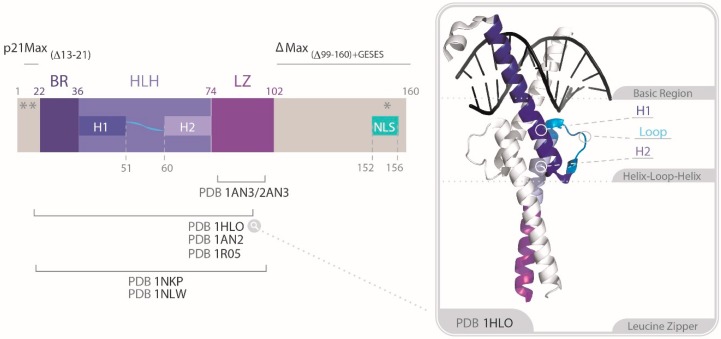
Schematic representation of the modular structure of p22MAX (MYC-associated factor X protein, UniProtKB code P61244). The naturally occurring (p21MAX) and oncogenic (DeltaMAX) MAX variants are identified. The three phosphorylation sites target of Casein Kinase II (CKII) are identified on MAX by the symbol * [38], both increasing the on- and off-rates of the homo- and heterodimers on DNA in vivo, are indicated by stars. The crystal structure of the DNA-bound p21MAX is displayed on the right. One monomer highlights the Basic Region in dark blue, the Helix-Loop-Helix domain in different shades of blue corresponding to H1, Loop and H2 elements and the Leucine Zipper in purple. The other monomer is shown in white while the double stranded DNA E-box is displayed in black.

**Figure 4 cells-09-01038-f004:**
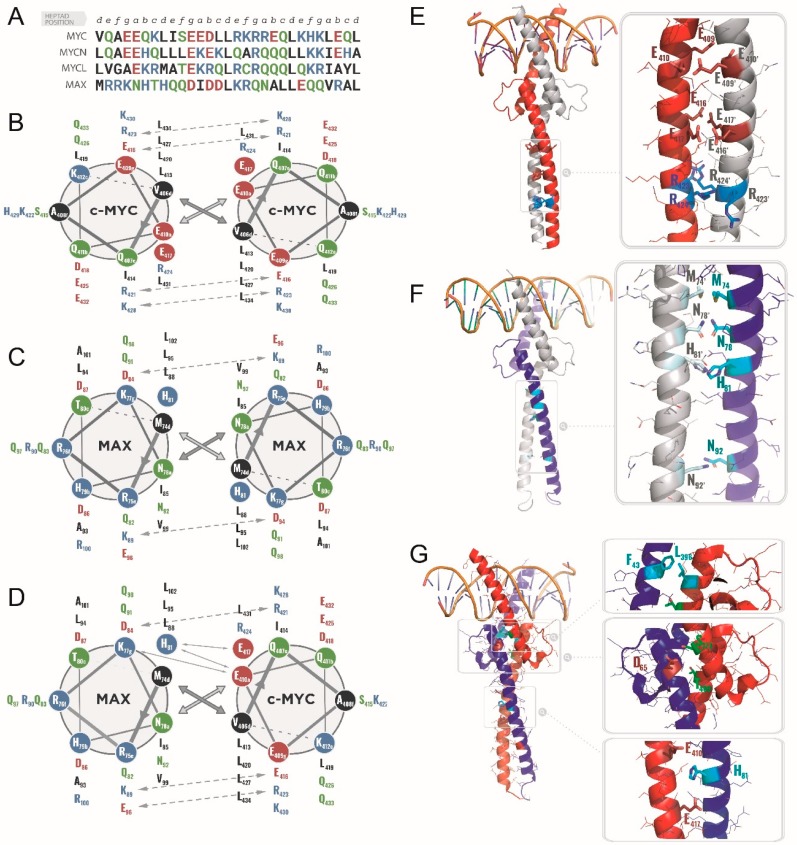
(**A**). Sequence alignment of the LZ of MYC, MYCL, MYCN, and MAX with heptad-repeat numbering. (**B**–**D**). Helical-wheel representation for the interfacial interactions in the LZ domains from MYC and MAX putative dimeric complexes: The MAX/MAX homodimer is shown in panel (**B**), the putative MYC/MYC homodimer in panel (**C**) and the MYC/MAX heterodimer in panel (**D**). The favorable and unfavorable inter-molecular contacts are displayed as arrows. The amino acids are colored blue for positively charge residues, red for negatively charged residues, green for polar residues and black for the remainder apolar residues. Adapted from Lavigne et al. 1995 [40]. (**E**,**F**) Molecular models of the tridimensional structure of a putative DNA-bound MYC homodimer (**E**), of the crystal structure of the MAX homodimer (PDB 1AN2, **F**) and of the MYC/MAX heterodimer (PDB 1NKP, **G**). (**E**). Homology model of a putative MYC/MYC homodimer (the sequence of MYC was superimposed on the MAX structure from 1NKP.pdb) with one monomer shown in red and the other in grey. The inset highlights the highly unfavorable electrostatic repulsions between the clusters of negatively charged interfacial residues Glu409-Glu410, Glu416-Glu417, and Arg423-Arg424 and their respective counterpart on the other monomer are shown with stick representation. Negatively charged residues are shown in red, positively charged residues in blue. F. Crystal structure of the MAX homodimer; the inset evidences the polar residues at interfacial a and d positions, namely Met74, Asn78, His81, Asn92 (shown in cyan stick representation). (**G**). Crystal structure of the MYC/MAX heterodimer with close up views on the MYC-Leu396/MAX-Phe43 interaction within the HLH (top inset); the Ser373 and Thr400 phosphorylation sites are shown in green (note: Asp65 from MAX, directly facing MYC-Ser373, is shown in red—middle inset); the interfacial, buried interaction between MAX-His81 (cyan) and MYC-Glu410 and Glu417 (in red) is shown (bottom inset).

**Figure 5 cells-09-01038-f005:**
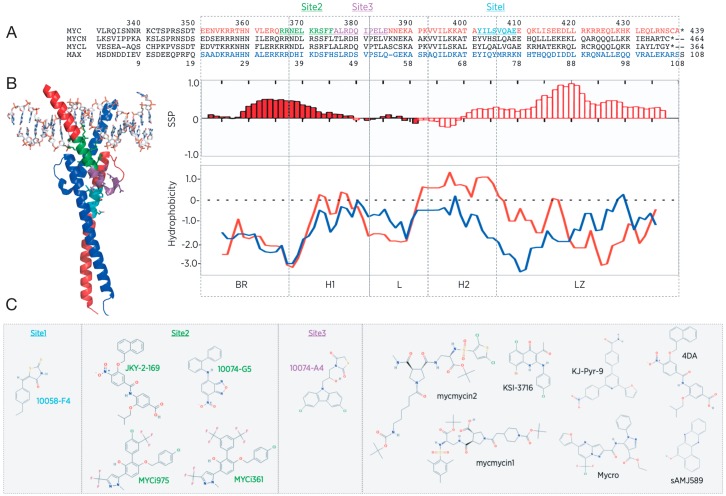
(**A**). Top: Sequence alignment of MYC (red), MYCN, MYCL, and MAX (blue). The binding sites for small molecules (SM) are identified: Site I in cyan, Site II in green and Site III in purple. The secondary structure prediction (SSP) for the MYC bHLHLZ in absence of MAX and of DNA is shown (adapted from Panova et al. [55]): Positive values correspond to helix and negative values indicate β–strand or polyproline II helix (PPII). Filled bars indicate predictions from the measured chemical shifts, while open bars refer to predictions based on signal broadening. Below, the hydrophobicity of the MYC (red) and MAX (blue) polypeptides as calculated from ProtScale using the Kyte and Doolittle reference values (https://web.expasy.org/protscale/pscale/Hphob.Doolittle.html). (**B**). Crystal structure of the DNA-bound MYC/MAX heterodimer (1NKP.pdb) highlighting the binding sites for SM (the same color code as in A is used). (**C**). Two-dimensional structure of the SM MYC inhibitors mentioned in the manuscript. The SM inhibitors binding to Site I, Site II or Site III are indicated.

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
