# Peer review of "Structural and Biophysical Insights into the Function of the Intrinsically Disordered Myc Oncoprotein"

_cells, 2020, doi:10.3390/cells9041038_

Round 1

Reviewer 1 Report

The submitted manuscript is an interesting and original review on the Myc oncogene. The review can be divided in two parts. The first is highly focused on the structural aspects of the Myc network. The second half discusses the state of the art on small molecules aimed to impair Myc activities. It covers decades of research performed on Myc. Many details are described including experimental aspects. In general, adding the images of the structures discussed in the text would help the reading. Similarly, including a table with the structures of the most relevant compounds would help, particularly in cases like of compounds 361 and that seem to bind the same region.

Minor points

I would not use the definition of infamous oncogene. All oncogenes when altered are infamous in a patient’s context. I would prefer important, key, critical, frequently altered ….

Some sections are too long. Perhaps the organization into subsections should help.

Page 3 lane 101. In fact in capital letter

Author Response

Please see the attachment or view here directly:

Comments and Suggestions for Authors

The submitted manuscript is an interesting and original review on the Myc oncogene. The review can be divided in two parts. The first is highly focused on the structural aspects of the Myc network. The second half discusses the state of the art on small molecules aimed to impair Myc activities. It covers decades of research performed on Myc. Many details are described including experimental aspects. In general, adding the images of the structures discussed in the text would help the reading. Similarly, including a table with the structures of the most relevant compounds would help, particularly in cases like of compounds 361 and that seem to bind the same region.

Reply:We thank reviewer 1 for his/her constructive suggestions. We have now included the structures discussed in the text in Figures 4 and 5 and hope this helps improving the quality of the manuscript.

Minor points

I would not use the definition of infamous oncogene. All oncogenes when altered are infamous in a patient’s context. I would prefer important, key, critical, frequently altered ….

Reply: We have removed the term “infamous” oncogene from the text.

Some sections are too long. Perhaps the organization into subsections should help.

Reply: We have reorganized the whole manuscript into subsections and have significantly shortened and rewritten some sections.

Page 3 lane 101. In fact in capital letter

Reply: We have corrected the typo (now line 123 in the new incarnation of the manuscript).

Reviewer 2 Report

Because new reviews from the MYC field seem to be published roughly once per month, I was not anticipating that the present review would provide much in the way of new insights. However, I was wrong. This review, by being strongly focused on the structural and biophysical aspects of the MYC oncoprotein, does provide a unique perspective on MYC structure and function that could only be otherwise achieved by spending a lot of time searching the literature related to structural studies. Therefore, I think that this review represents a worthy contribution to the extant literature on MYC.

Nonetheless, there are some problems. It is exhaustive in its coverage of the literature but also exhausting to read. It is very very dense and many of the sections are quite long. In general I would suggest that the authors organize the text into subsections so that readers can get a better idea of how to navigate the review and also find their way to subjects that they are particularly interested in. In addition, the authors should consider a way to provide a larger image of the Myc-Max heterodimer structure in which they can actually show the structures and regions which they discuss in the review, e.g. the basic region-helix 1 border , the loop, etc. I suspect this would be more useful then showing the helical wheels in figure 4.

Some specific comments (major and minor):

Line 28 - This sentence mentions the MYC family paralogs but it gives the mistaken impression that MYCN and MYCL are only associated with neuroblastoma and lung cancers respectively - this is confusing and misleading.

Line 35 and the Fig. 1 timeline-despite what the text says, the discovery of Max is not indicated in Fig. 1 - it should be in your timeline. You might consider also including in the text and in fig 1 the conceptually important discovery by two groups that Myc is predominantly localized in the cell nucleus - this was the first retroviral oncoprotein to be found in the nucleus.

Line 50 - it’s histone deacetylases not “histones deacetylase”. It would be useful to have a native English speaker proofread the MS.

Line 56 – Yes, MYC has a relatively short half-life but what makes it “notorious”?

Line 76 - a bit confusing- the defining feature of the Myc Boxes is that that are evolutionarily highly conserved regions.

Line 83 - might be useful to stress that 439 amino acids is the length for human c-MYC.

Line 85 – The idea that a MYC inhibitor would only have efficacy if it blocked all three MYC paralogs would only be correct if inhibition of one paralog leads to increased expression of the other paralogs? In many cancers inhibition of a single MYC family member is sufficient to block growth.

Line 216: The authors might consider citing the recent paper by Mathsyaraja et al (Genes Dev 2019) who demonstrate that deletion of MAX results in very rapid degradation of MYC.

Line 256 – It’s not obvious why a faster dimerization rate would decrease speed of binding to DNA. Please explain.

Line 400 – Weird sentence. An “interaction was found to bind…”?

Line 414- Where is the reference for BIN1 function? I don’t think the biology of BIN1 has been well established. I checked the recent comprehensive MYC interactome paper from the Penn lab – they apparently did not find that BIN1 was one of the many proteins that bind Myc.

Line 686 – The authors  attempt to generalize about the difficulty in finding MYC targets and argue that there is no common set of MYC targets in multiple cell types. However ribosomal proteins, ribosomal RNA, tRNA synthases and other growth related genes are pretty close to universal targets of MYC from Drosophila to mammals. This idea is raised later in the review but should be mentioned here. Maybe the really important point is that Myc targeting is cell type and cell context dependent.

Lin e703 – Define EC50

Line 721 – define coacervate

Line 727 ref 165  - this is a review citing another review. The original reference for this (ref 96) should be cited here.

Line 745 – Here is a sentence worthy of James Joyce:

In such chain of events, one important rate-limiting step is the formation/expansion of the liquid condensates or coacervates that depends on the rapid and  enthalpy-driven binding of MYC/MAX to DNA, which in turn provides an entropic drive and change  in the local chemical potential that stimulates the partition of the coregulators of transcription and  post-translational modifiers according to their sequence-specified solubility for each phase.

This sentence could easily be reduced to two or three sentences that would not give the reader a headache. There are quite a few sentences like this in the review. Careful proofreading would strengthen the text.

Reviewer 3 Report

This is a comprehensive and detailed review of the subject presented in a very systematic manner. The authors made a great effort to dissect the complex subject into various segments. The arguments are supported by well-referenced scientific work. 

My only comment is to include some perspective as well as a position by authors on the following:

1- What is the current approach towards targeted therapies mitigating high MYC expression in cancer and why direct inhibition has been unsuccessful.

2- If the current state of knowledge, provide any insight to adopt futuristic venues in exploration for targeting MYC directly in cancer therapy.  
